# Lithium Chloride Sensitivity in Yeast and Regulation of Translation

**DOI:** 10.3390/ijms21165730

**Published:** 2020-08-10

**Authors:** Maryam Hajikarimlou, Kathryn Hunt, Grace Kirby, Sarah Takallou, Sasi Kumar Jagadeesan, Katayoun Omidi, Mohsen Hooshyar, Daniel Burnside, Houman Moteshareie, Mohan Babu, Myron Smith, Martin Holcik, Bahram Samanfar, Ashkan Golshani

**Affiliations:** 1Department of Biology and Ottawa Institute of Systems Biology, Carleton University, Ottawa, ON K1S 5B6, Canada; Maryamhajikarimlou@cmail.carleton.ca (M.H.); KathrynHunt3@cmail.carleton.ca (K.H.); GraceKirby@cmail.carleton.ca (G.K.); SaraTakalloo@cmail.carleton.ca (S.T.); sasikumarjagadeesan@cmail.carleton.ca (S.K.J.); KatayounOmidi@cmail.carleton.ca (K.O.); Mohsen.Hooshyar@Carleton.ca (M.H.); daniel.burnside@carleton.ca (D.B.); houman.moteshareie@carleton.ca (H.M.); MyronSmith@cunet.carleton.ca (M.S.); bahram.samanfar@canada.ca (B.S.); 2Department of Biochemistry, Research and Innovation Centre, University of Regina, Regina, SK S4S 0A2, Canada; mohan.babu@uregina.ca; 3Department of Health Sciences, Carleton University, Ottawa, ON K1S 5B6, Canada; MartinHolcik@cunet.carleton.ca; 4Agriculture and Agri-Food Canada, Ottawa Research and Development Centre (ORDC), Ottawa, ON K1Y 4X2, Canada

**Keywords:** lithium chloride, mode of activity, toxicity, bipolar disorder, side effects, gene expression, yeast, translation

## Abstract

For decades, lithium chloride (LiCl) has been used as a treatment option for those living with bipolar disorder (BD). As a result, many studies have been conducted to examine its mode of action, toxicity, and downstream cellular responses. We know that LiCl is able to affect cell signaling and signaling transduction pathways through protein kinase C and glycogen synthase kinase-3, which are considered to be important in regulating gene expression at the translational level. However, additional downstream effects require further investigation, especially in translation pathway. In yeast, LiCl treatment affects the expression, and thus the activity, of *PGM2*, a phosphoglucomutase involved in sugar metabolism. Inhibition of *PGM2* leads to the accumulation of intermediate metabolites of galactose metabolism causing cell toxicity. However, it is not fully understood how LiCl affects gene expression in this matter. In this study, we identified three genes, *NAM7*, *PUS2*, and *RPL27B*, which increase yeast LiCl sensitivity when deleted. We further demonstrate that *NAM7*, *PUS2*, and *RPL27B* influence translation and exert their activity through the 5′-Untranslated region (5′-UTR) of *PGM2* mRNA in yeast.

## 1. Introduction

Bipolar disorder (BD) is known to be associated with dysregulated signaling pathways. For the past few decades lithium chloride (LiCl) has been a key treatment option for those living with BD, because of its neuroprotective effects [1,2]. It has been used as a psychotropic drug and has anti-suicidal and anti-depressant effects in BD patients [3,4] by possibly targeting miRNAs regulating proteins involved in mood stability [5]. LiCl plays a role in two key signal transduction pathways including Protein Kinase C and Glycogen Synthase Kinase-3, both of which are involved in neurodevelopment and neuronal plasticity [1]. It has been shown to have neuroprotection effects, protecting neurons against apoptosis and disturbing other pathways including cell proliferation, apoptosis, cell fate, and translation [5,6,7,8].

As a monovalent cation which targets the nervous system, LiCl has been studied as a potential treatment for Alzheimer’s disease (AD), Parkinson’s disease, and Huntington’s disease [7,9]. Increased plaques and tangles in the dopaminergic regions leads to increased cell death, a well-known characteristic of AD [1]. The efficacy of LiCl as a treatment for AD appears to be linked to LiCl’s ability to decrease levels of amyloid β and decrease phosphorylation of the tau protein [10]. Although much is known about the activity of LiCl, it is still unclear how it affects the cell at the molecular level. The influence of LiCl on gene expression and its secondary effects are also not well-understood. It has been observed that, after long term treatment of BD patients with LiCl, the expression of genes involved in the phosphoinositide/Protein Kinase C (PI/PKC) signaling cascade are disturbed. As a result, this causes disruption in function of nerve cells [7,11,12,13], renal cells, and liver cell damages [2]. Understanding how LiCl is regulating the expression of specific genes related to these pathways, as well as others, requires further investigation.

Yeast is particularly sensitive to LiCl when galactose is used as a sugar source. This is because LiCl alters the expression and activity of *PGM2*, a phosphoglucomutase which mediates the entry of galactose into glycolysis [14,15]. *PGM2* also converts glucose-1-P to glucose-6-P and, if inhibited, it leads to accumulation of intermediate metabolites causing toxicity in yeast cells [16]. When LiCl is added to galactose media, yeast cell growth is severely reduced due to impaired glycolysis. LiCl also reduces the levels of uridine diphosphate glucose (UDP-glucose) and disrupts associated pathways in the presence of glucose. Other studies have also suggested that LiCl might inhibit RNA processing enzymes and rapidly reduce expression of ribosomal protein genes [17,18]. This results in a decreased number of mature mRNAs in the cytoplasm [17], suggesting inhibition at translational level.

Sensitivity of yeast to LiCl has also been connected to transport kinetics and cellular metabolism [19,20]. Other studies have reported that LiCl effects sporulation in yeast by delaying the process [19]. Zhao et al. [20] found 25 genes involved in sporulation and 35 gene in membrane docking and fusion that, when deleted, yeast showed increased sensitivity to LiCl. Among them was *HAL1* gene that encodes a halotolerant protein kinase known to be crucial for integrity of vacuolar protein sorting pathway (VPS). In the same study, it was shown that LiCl can also affect ion homeostasis. On glucose media, addition of 100 mM LiCl reduced transcription rate in yeast [19] and in *Candida albicans* [21]. It is proposed that the sensitivity of *C. albicans* deletion mutant for *MAF1*, a repressor of RNA polymerase III might be due to LiCl targeting transcription [21]. Effect of LiCl on inositol inhibition in yeast was investigated by Lopez et al. [22]. The authors showed that the yeast inositol monophosphatase is targeted by LiCl, affecting calcium signaling and inositol biosynthesis [22].

Other studies suggest that LiCl inhibits the initial steps of protein synthesis. One study has demonstrated that LiCl may be able to disrupt the association between the translation initiation factor eIF4A, an RNA helicase, and the rest of the translational machinery, resulting in impaired translation initiation [14]. However, this was not observed in yeast growing on glucose media. In the same study, it was shown that the over-expression of eIF4A was able to revert the inhibition of translation due to LiCl [14]. In a recent investigation, we showed a connection between LiCl and translation of structured mRNAs [23].

Large scale sensitivity analysis of yeast for other compounds including cobalt and lead has also been investigated [24,25]. It was reported that yeast sensitivity to cobalt is likely because of the effect of cobalt on oxidative stress response and unfold protein response pathways [25]. In another study, it was shown that lead and cadmium seem to target the metabolism and cellular transport processes in yeast [24].

In the current study, we observed that the deletion of three yeast genes, *NAM7*, *PUS2*, and *RPL27B* increased the sensitivity of yeast cells to LiCl. These genes have never been studied in cell responses linked to LiCl. *NAM7* is an important gene involved in efficient termination of translation, especially in nonsense-mediated mRNA decay. This gene codes for an RNA helicase, which binds to the small ribosomal subunit, controlling decay for a specific subset of mRNAs. *PUS2* is involved in tRNA modification in the mitochondria and also pseudouridylation of some nuclear mRNAs. And lastly, *RPL27B* encodes for the large ribosomal subunit, but it is not known if it is involved in translation regulation. In this study, we demonstrate that deletion of yeast genes *NAM7*, *PUS2*, and *RPL27B* increases sensitivity of yeast to LiCl. Follow-up genetic investigations suggest involvement of these genes in yeast LiCl sensitivity through their effect on translation of *PGM2*.

## 2. Results

### 2.1. Deletion of NAM7, PUS2, and RPL27B Increases Yeast Sensitivity to Lithium Chloride

Chemical sensitivity of mutant strains under treatment of a target compound is an investigation tool that allows us to determine the targeted pathways of the compound and allows observation into the effects on the cell at a molecular level [26,27,28]. While investigating yeast gene deletion mutants that are sensitive to LiCl, we identified three gene deletion mutants: *NAM7*, *PUS2*, and *RPL27B* which showed increased sensitivity to LiCl (Figure 1B), compared to control media without LiCl treatment (Figure 1A). In the spot test assay, we demonstrate that deletion of *TIF2* (eIF4A) causes increased sensitivity to 10 mM LiCl when grown in media with galactose as the carbon source. Deletion of *NAM7*, *PUS2*, and *RPL27B* showed dramatic growth reduction in the same manner (Figure 1B), suggesting sensitivity to LiCl. Reintroduction of the deleted genes back into the chromosome of the corresponding gene deletion mutants reversed this growth reduction, further connecting the observed phenotypes to the deleted gene (Appendix A). We also tested to see the sensitivity of yeast strains to LiCl using glucose as the carbon source; however, no significant toxicity was observed (data is not shown).

It has been previously shown that over-expression of *TIF2* reverts the toxicity of sensitive strains to LiCl [14]. To see if our candidate genes would revert LiCl sensitivity in the same manner, we transferred over-expression plasmid of our candidate genes into corresponding deletion strains spotted on media containing 10 mM LiCl. When the plasmids were introduced to the mutants, the fitness of the strains was recovered, proposing they may have similar function as eIF4A in the cell (Figure 1C). The molecular activity of *NAM7*, *PUS2*, and *RPL27B* has never been connected to LiCl sensitivity and the molecular pathways related to that, making them interesting targets for further investigation.

These findings were confirmed with colony count measurement analysis, providing a quantitative approach (Figure 2). In this experiment, the number of colonies seen in the presence of LiCl in the media is compared and normalized to the number of colonies seen in control media and wild type (WT). Using this data, we were able to calculate the decreased percentage of colonies seen in deletion strains. As seen in Figure 2, deletion of *NAM7*, *PUS2*, *RPL27B***,** or *TIF2* lead to decreased colony formation.

LiCl targets *PGM2* expression which leads to an accumulation of galactose intermediate metabolites, which becomes toxic for yeast cells including galactose-1-p [14,15,29]. In yeast, *GAL1* is the galactokinase that phosphorylates α-d-galactose to α-d-galactose-1-phosphate in the first step of galactose catabolism. The *PGM2* enzyme facilitates the entry of galactose into glycolysis and converts glucose-1-phosphate to glucose-6-phosphate. To investigate the influence of *NAM7*, *PUS2*, and *RPL27B* on LiCl toxicity through galactose metabolism, we generated double gene deletions for *NAM7*, *PUS2*, and *RPL27B* with the *GAL1* gene. We observed that when *GAL1* was deleted, double mutant cells did not show sensitivity to LiCl treatment (Figure 1B).

### 2.2. NAM7, PUS2, and RPL27B Regulate the Expression of PGM2 at the Level of Translation

Since altered expression of *PGM2* has been seen in the presence of LiCl [29], we examined to see if *NAM7*, *PUS2*, or *RPL27B* would affect *PGM2* expression at the translation level. Pgm2p was tagged with GFP and western blot analysis was done to see if any of the three genes altered *PGM2* expression (Figure 3A). When cells were exposed to 10 mM LiCl, deletion of any three of the genes resulted in reduced protein levels of Pgm2p compared to WT. When cells were not exposed to LiCl, deletion of *NAM7*, *PUS2*, or *RPL27B* showed no significant difference in cell fitness.

To see the impact of *NAM7*, *PUS2*, or *RPL27B* on *PGM2* transcription, qRT-PCR was performed to measure the mRNA content of *PGM2*. As seen in Figure 3B, deletion of *NAM7*, *PUS2*, or *RPL27B* did not significantly alter *PGM2* mRNA levels on control media compared to WT. The mRNA content of *PGM2* increased in cells treated with LiCl but showed no significant difference between mutant strains and WT. This suggests that these genes do not impact *PGM2* at transcriptional level. Thus, this would imply that *NAM7*, *PUS2*, or *RPL27B* affect expression of Pgm2p at the protein level. These results are in agreement with a previous study by Sofola-Adesakin [7], who demonstrated that LiCl impaired gene expression during protein synthesis and not during transcription [7].

### 2.3. Translation of β-Galactosidase Reporter mRNA with a Hairpin Structure Is Altered by Deletion of NAM7, PUS2, and RPL27B

The 5′-UTR of *PGM2* mRNA has a highly structured region [30] (Appendix A). *PGM2* expression is severely reduced in the absence of translation initiation helicase *TIF2*, a protein responsible for unwinding mRNA structures during translation [14]. Since we observed that *NAM7*, *PUS2*, and *RPL27B* are likely to function in the translation pathway, we decided to examine if these genes are possibly impacting translation of highly structured mRNAs. For this experiment, we inserted 5′-UTR of *PGM2* in front of a *LacZ* expression cassette of p416 plasmid (pPGM2). From here, pPGM2 was then transformed into deletion mutant strains of our candidate genes and WT. *β-galactosidase* activity was measured as a reference for translation activity (Figure 4A). It was shown that translation activity significantly decreased in *Nam7**Δ*, *Pus2**Δ*, and *Rpl27b**Δ* carrying plasmid with hairpin when compared to translation activity with control plasmid (Figure 4B).

To investigate whether this effect is specific to the 5′-UTR of *PGM2* mRNA, we used a second construct with a strong hairpin structure on its 5′-UTR (p281-4) [31]. As a control, we used p281, which lacks a secondary structure [31]. We observed that *β-galactosidase* activity was reduced in p281-4 when *NAM7*, *PUS2*, or *RPL27B* was deleted, whereas, for p281, there was no significant difference between the mutants and WT (Figure 5A,B). These results demonstrate that the tested genes do not appear to affect translation of mRNAs lacking structured regions and exert their activities specifically on structured mRNAs.

Since *NAM7*, *PUS2*, and *RPL27B* impacted translation of structured reporter mRNAs, we wanted to see if they were able to impact translation of other naturally structured mRNAs compared to the previous ones which were designed by software. For this purpose, we designed two *β-galactosidase* mRNA reporters with different complex RNA structures: pTAR and pRTN. pTAR has the 5′-UTR of the *HIV-tar1* gene, which has a strong hairpin loop, while pRTN has the 5′-UTR of the *FOAP*-*11* gene, which has a highly structured region [32,33]. When *NAM7*, *PUS2*, or *RPL27B* were deleted, levels of *β-galactosidase* expression were significantly reduced (Figure 5C,D).

### 2.4. Genetic Interaction Analysis Further Connects the Activity of NAM7, PUS2, and RPL27B to Protein Biosynthesis

Genetic interaction (GI) analysis assumes that parallel pathways allow for plasticity and tolerance against random deleterious mutations, protecting cells and maintaining cell homeostasis if one gene is deleted or mutated in a pathway [34]. This means that a gene in one pathway can compensate for the lack of gene activity in a parallel pathway, allowing the cell to survive. Accordingly, when two genes in parallel pathways are deleted, cell fitness decreases (sickness) or the cell dies (lethality). As a result, we assert that the two genes are genetic interactors, or in other words, they are functionally working in parallel pathways. These aggravating interactions are also known as negative genetic interactions (nGIs). nGIs are useful in many studies to understand gene function and pathway crosstalk [26,35,36].

Analysis of GIs in yeast is done by mating two types of yeast: α-mating type (Mat α), and a-mating type (Mat a). Mat “α” carries the target gene deletion, which is crossed with Mat “a”, an array of single gene deletions to produce double gene deletions [37]. Fitness of double deletions is measured using colony size assessment [38]. Using this method, we made double deletions for each of our three query genes *NAM7*, *PUS2*, and *RPL27B* with nearly 1000 other genes to screen for possible genetic interactions. This experiment included a random set of 304 genes as controls.

We observed several interesting nGIs with *NAM7*, including *PRP22*, *TIF2*, *GCD11*, and *PRT1. PRP22* is a DEAH-box RNA helicase, *TIF2* codes for the translation initiation factor eIF4A, *GCD11* forms part of the small subunit of eIF2, and *PRT1* is the subunit of eIf3. Other interacting genes involved in translation initiation are *YGR054W* and *HCR1* (Figure 6).

*PUS2* interacted with *DHH1*, *EAP1*, and *HCR1* among others. *DHH1* codes for an ATP-dependent RNA helicase, *EAP1* codes for an eIF4E-associated protein, and *HCR1* codes for a subunit of the eIF3 translation initiation factor, which is also important in binding of initiation factors to 40S subunit and AUG recognition along with *HCR1* as an RNA recognition motif [39]. Many of *PUS2* nGIs were involved with ribosomal structural proteins (Figure 6).

As expected, *RPL27B* interacted with a number of translation genes (Figure 6). This included *DHH1*, *EAP1*, *SLH1*, and *SKI2*, which have RNA helicase activity, as well as *ECM32*, which has DNA helicase activity and is also involved in modulating translation termination.

Comparing the nGI profiles for *NAM7*, *PUS2*, and *RPL27B* we noticed some interesting common hits. *HCR1* is a subunit of eIF3 and is known to be important for scanning efficiency specially in cooperation with *DED1* (RNA helicase) on long 5′-UTRs [40] and binding with *DHX29* in human cells [41]. *PRT1* is also crucial in recognition of the right start codon. During scanning, *PRT1* inhibits leaky scanning by promoting the stability of ribosomes on mRNAs possibly by changing its conformation [42,43]. *XRN1* is known to be involved in mRNA decay and transcription regulation, but recently it was proposed that it might also play a role in translation pathway by regulating translation of specific mRNAs through binding to eIF4F complex, a translation initiation complex [44]. *DED1* is an ATP-dependent helicase that associates with eIF4A to regulate translation initiation [45,46]. Methylation of *DED1* strengthens its binding to eIF4A and to *XRN1* [47], suggesting its potential effect on translation regulation. *XRN1* was also proposed to interact with another helicase, *DHH1*, in yeast to control translation by decapping mRNAs for degradation [48,49]. *P54* (homologue of *DED1* in humans) was shown to be important in localization and assembly of P-bodies in the cell [48].

The comparison of this data to the published nGIs in *Saccharomyces* Genome Database (SGD) shows an expected degree of overlap. For example, for *NAM7*, except for an observed nGI with *YGR054W* in the current study, other observed nGIs overlap with the reported interactions. This provides further confidence for the validity of our observations.

Another form of nGIs, called conditional nGI, can be studied under sub-inhibitory concentration of chemicals. It is used to investigate possible candidate gene functions that are activated under a specific condition [27,50]. For this experiment, we studied nGIs for *NAM7*, *PUS2*, and *RPL27B* under a mild sub-inhibitory concentration of LiCl (3 mM). Shown in Figure 7, new nGIs were observed for our candidate genes. The new interactions hinted to new conditional functions for *NAM7*, *PUS2*, and *RPL27B* in regulation of translation. We observed a number of common interactors between the three query genes including *BCK1*, *CTK1*, and *MCK1*. *BCK1* is important in negative regulation of translation under stress conditions. It is involved in deadenylation and decapping of mRNAs to be degraded in connection with *DHH1* [51,52,53,54]. On the other hand, *CTK1* is a conserved kinase that phosphorylates RPS2p, one of the components of small ribosomal subunit. It affects translation fidelity during elongation, as well as phosphorylation of other translation initiation factors, including eIF4A, eIF5, eIF4G, and eIF3 [55,56]. *MCK1* is involved in phosphorylation-dependent protein degradation, among other roles.

Phenotypic Suppression Array (PSA) analysis focuses on another form of interaction where over-expression of one gene compensates for the absence of another gene, under a drug treatment [26,36,57,58]. Here, we treated the arrays of mutant strains with 10 mM LiCl, and, consequently, some showed sensitivity. Then, we were able to revert LiCl sensitivity in a number of deletions by introducing over-expression plasmid of *NAM7*, *PUS2*, and *RPL27B*. Interestingly, over-expression of *NAM7*, *PUS2*, and *RPL27B* compensated for the sensitivity of four mutual gene deletions *ITT1Δ*, eIF2A*Δ*, *EAP1Δ*, and *PSK2Δ* (Figure 8). *Eap1* encodes for an eIF4E-associated protein that accelerates decapping of mRNA and negatively regulates translation [53]. *ITT1* interacts with the translation release factor eRF3 and modulates the efficiency of translation termination. eRF3 is not only important in regulation of translation termination and cell cycle regulation but also is shown to mediate mRNA decay through interaction with UPF family of proteins that include one of our query genes *NAM7* [59]. Itt1p interacts with Mtt1p, an RNA helicase and poly(A) binding protein Pab1p involved in mRNA circularization and ribosome recycling [59,60]. eIF2A deletion also showed sensitivity to 10 mM LiCl and was reverted by over-expression of *NAM7*, *PUS2*, and *RPL27B* in our experiments. eIF2A, a translation initiation factor, is shown to regulate IRES-mediated translation and because of its physical interaction with Ded1p and eIF4A, it has been suggested to play a role in translation regulation of certain mRNAs [61]. *PSK2* is known to be important in regulating both sugar metabolism and translation through phosphorylation of intermediate molecules [62]. In addition, *PSK2* is shown to function as a trans-acting factor for translational regulation of mRNAs with upper Open Reading Frames (uORFs), including *ROK1* that encodes an RNA helicase [63].

## 3. Discussion

Chemical genetic profiling is an investigating tool that enables us to determine not only the genetic targets of a compound, but also the effect of the compound on different genes and pathways. In addition, it can be used as a powerful tool to identify new gene functions [27,28,64,65]. Complementary assays can be used to generate the chemical-genetic profiles of various compounds. In the current stud y, we used the yeast nonessential gene knockout library to investigate LiCl sensitivity. A more inclusive collection of bar-coded yeast strains containing both essential and nonessential genes with decreased abundance of mRNA perturbation (DAmP) is also available to study chemical genetics [66]. Using this approach, it was shown that the protein product of *IPP1* genes, a cytoplasmic inorganic pyrophosphatase is specifically inhibited by NaF [67].

LiCl has been shown to have a broad effect on the expression of different genes and pathways when used as a therapeutic compound, but its exact mechanism of function is not fully understood. Here, we investigated yeast gene deletion mutant strains for their sensitivity to LiCl treatment and identified three genes, *NAM7*, *PUS2*, and *RPL27B* that when deleted, their corresponding deletion mutant strains showed increased sensitivity to LiCl. It has been shown that the over-expression of *TIF2* that codes for eIF4A, the helicase enzyme involved in translation initiation step, reverts LiCl sensitivity in yeast cells growing in galactose media [14]. Here, we observed that the over-expression of our candidate genes confer similar phenotypes to that for *TIF2*, suggesting a linked activity for these gene. The molecular activity of *NAM7*, *PUS2*, and *RPL27B* has not been connected to LiCl sensitivity and the molecular pathways related to that, making them interesting targets for investigation. Yeast cells are sensitive to LiCl treatment due to the accumulation of the galactose intermediate metabolites including galactose-1-p [14,15]. It is suggested that LiCl targets *PGM2*, the phosphoglucomutase enzyme, and prevents conversion of glucose-1-phosphate to glucose-6-phosphate by changing its expression [29]. In the current study, we show that *NAM7*, *PUS2*, and *RPL27B* affect *PGM2* expression. More specifically, we provide evidence that *NAM7*, *PUS2*, and *RPL27B* affect the translation of *PGM2* and different reporter *β-galactosidase* mRNAs and that they exert this activity through the structured 5′-UTR regions of the mRNAs. Very recently we reported on the new activity of two additional genes, *YTA6* and *YPR096C* on the translation of structured mRNAs [23]. Identification of new gene functions that affects this process suggests that the translation of structured mRNAs might be more complicated than originally thought. It appears that different factors can contribute to the translation of these mRNAs and that structured on mRNAs are subjected a more complex regulator network.

The effect of *NAM7*, *PUS2*, and *RPL27B* on the translation of structured mRNAs can be explained in different ways. The simplest explanation is that one or more of these factors might have mRNA helicase activities, which can directly contribute to the unwinding of the structured regions of the mRNA. *NAM7* is previously reported to have a helicase activity. Therefore, it is likely that the observed activity of *NAM7* in the current study might be linked to its reported helicase function. No helicase activities for *PUS2* and *RPL27B* has been previously reported making the direct unwinding of mRNA structures a less likely explanation for their activities. It is possible that one or more of these factors could influence the activity of other complexes with helicase activity. In this fashion they can play an accessory role in the translation of structured mRNAs. *RPL27B* codes for a ribosomal protein. A possibility is that the ribosomes in the presence of this factor can translate structured mRNAs more efficiently. Similarly, these factors could modify the activity of other helicases including eIF4A that is known to unwind the structured regions of mRNAs during translation initiation. An alternative explanation is that one or more of these factors may function to recruit other helicases to the mRNA. Pus2p is known to have an RNA binding and modification activity, among others. A possible explanation is that Pus2p may bind/modify mRNAs and by doing so it mediates the recruitment and/or binding of other factors to that region. The mechanisms by which these factors can influence the translation of structured mRNAs merits further investigation.

The connection between LiCl and translation of structured yeast mRNAs observed in the current study may aid in our improved understanding of the BD treatment. It is generally thought that LiCl has a neuroprotection effect, protecting neurons against apoptosis. In light of the current findings, it is possible that LiCl exerts its activity by altering the translation of structured mRNAs involved in apoptosis. In future, it would be interesting to investigate the activity of the structured mRNAs involved in apoptosis of the neuron cells for their involvement in the BD progression. It is possible that the expression of these mRNAs may provide an effective target against BD.

## 4. Materials and Methods

### 4.1. Strains, Plasmids, Gene Collections, and Cell and DNA Manipulations

MAT “a” mating strain Y4741 orfΔ::KanMAX4 his3Δ1 leu2Δ0 met15Δ0 ura3Δ0 and MAT “α” mating strain, Y7092 can1Δ::STE2pr-Sp_his5 lyp1Δ his3Δ1 leu21Δ0 ura3Δ0 met15Δ0 were used. Deleted mutant strains were obtained from yeast non-essential gene knockout mutant library (YKO) [68]. This library was also used for Synthetic Genetic Analysis (SGA) to make double mutants. Over-expression plasmid of candidate genes were extracted from yeast over-expression collection [69], and yeast PGM2-GFP fusion strain was purchased from Yeast GFP Clone Collection from Thermofisher^®^, Ottawa, Canada which was used for qPCR and western analysis purposes as described previously [26,27,35]. The integrity of this strain was confirmed using PCR and chemical sensitivity analyses. Yeast gene deletions in MAT “α” strain were performed by PCR transformation and homologous recombination through Lithium Acetate method as described in previous experiments [65,70]. Reintroduction of the deleted genes back into their corresponding deletion mutant strains was performed using a modified SGA method and random spore analysis [37,71] by mating the Mat “a” deletion mutant strain, carrying a G418 resistance gene marker in place of the deleted gene with a WT Mat “α” strain carrying a nourseothricin sulfate (clonNAT). In brief, after the sporulation step, spores were allowed to germinate on media containing canavanine and thialysine, to select for MAT “a” haploid progeny. Cells were then transferred to media containing clonNat. Colonies formed from individual cells were tested for sensitivity to G418 using replica plating. Those resistant to clonNAT but sensitive to G418 were selected and tested using PCR analysis to confirm the presence of the target gene in their chromosomes.

p281 construct, carrying a *LacZ* expression cassette under the control of a gal promoter, was used as our control plasmid. p281-4 with a strong hairpin (5′-GATCCTAGGATCCTAGGATCCTAGG ATCCTAG-3′), inserted upstream of *LacZ* expression cassette was used as our investigating construct [31]. In order to identify successful transformation in gene knockout experiments, the pAG25 plasmid was used as a DNA template in PCR reactions to amplify a clonNAT resistance gene marker. All plasmids carried an ampicillin resistance gene which was used as a selection marker in *E. coli* DH5α, and a URA3 marker gene for selection in yeast. Kanamycin and NAT drug was used as selectable markers for gene knock out in Mat “a” and Mat “α”, respectively.

p416 construct carries a *LacZ* expression cassette under the transcriptional control of a gal promoter which was used as our plasmid control in translation activity assays. To generate reporter *β-galactosidase* mRNAs under the translational control of complex RNA structures, three different fragments were cloned upstream of the *LacZ* expression cassette in p416 construct, using the XbaI restriction site. Using this method, three expression constructs were designed as follows: pPGM2 construct which contains the 5′-UTR of *PGM2* gene (5′TAATAAGAAAAAGATCAC CAATCTTTCTCAGTAAAAAAAGAACAAAAGTTAACATAACAT 3′), pTAR construct which contains the 5′-UTR of HIV-tar1 gene (5′ GGGTTCTCTGGTTAGCCAGATCTGAGCCCGGGAGCT CTCTGGCTAGCTAGGGAACCCACTGCTTAAGCCTCAATAAAGCTTGCCTTGAGTGCTTCAAGTAGTGTGTGCC 3′) and pRTN which contains the 5′-UTR of FOAP-11 gene (5′ GGGATTTTTACA TCGTCTTGGTAAAGGCGTGTGACCCATAGGTTTTTTAGATCAAACACGTCTTTACAAAGGTGATCTAAGTATCTC 3′).

YP (1% Yeast extract, 2% Peptone) or SC (Synthetic Complete) with selective amino acids (0.67% Yeast nitrogen base w/o amino acids, 0.2% Dropout mix) either with 2% dextrose or 2% galactose as a carbon source were used as culture medium for yeast and LB (Lysogeny Broth) was used for *E. coli* cultures. Two percent agar was used for all solid media. Yeast plasmid extraction was performed using yeast plasmid miniprep kit (Omega Biotek^®^, Norcross, GA, USA) and *E. coli* plasmid extraction was carried out using GeneJET plasmid miniprep kit (Thermofisher^®^, Ottawa, ON, Canada and Bio-Basics^®^, Toronto, ON, Canada) according to the manufacturers’ instructions.

### 4.2. Chemical Sensitivity Analysis

For Chemical sensitivity analysis, yeast cells were grown from independent colonies to saturation for two days at 30 °C in liquid YPgal. Following incubation, four serial dilutions (10^−1^ to 10^−4^) of the cell cultures were spotted onto solid media with or without LiCl treatment. Different concentration of LiCl (10 mM and 3 mM) were tested for growth sensitivity in media containing galactose or glucose as described previously [15,29]. After 48 h, the sensitivity was determined based on colony size and number [65].

For quantification analysis, 100 µL of diluted (10^−4^) oversaturated cell culture, as described above, was spread on YPgal plates with or without LiCl treatment. After incubation at 30 °C for two days, plates were analyzed based on colony numbers. Each experiment was repeated at least three times. *T*-test analysis (*p*-value ≤ 0.05) was used to determine statistically significant results compared to WT control.

### 4.3. Quantitative β-Galactosidase Assay

The effect of 5′-UTR regions to mediate translation in different yeast strains was examined using *LacZ* reporter systems. This assay was used to evaluate activity of *LacZ* cassette in different mutant strains as a reference for translation activity. Quantitative *β-galactosidase* assay was performed using ONPG (O-nitrophenyl-α-d-galactopyranoside) as described [72,73]. Each experiment was repeated at least three times.

### 4.4. Quantitative Real Time PCR (qRT-PCR)

To determine mRNA level of *PGM2* in different deletion mutants, *PGM2*-GFP yeast strain were grown in YPgal overnight with or without LiCl treatment. Total RNA was extracted with Qiagen^®^ RNeasy Mini Kit. Complementary DNA (cDNA) was made using iScript Select cDNA Synthesis Kit (Bio-Rad^®^, Mississauga, ON, Canada) according to the manufacturer’s instructions. cDNA was then used as a template for quantitative PCR. qPCR was carried out using Bio-Rad^®^ iQ SYBR Green Supermix and the CFX connect real time system (Bio-Rad^®^), according to the manufacturers’ instructions. *PGK1* was used as a constitutive housekeeping gene (internal control). The procedure and data analysis were performed according to MIQE guidelines [74].

The procedure was done in three repeats and *t*-test analysis (*p*-value ≤ 0.05) was used to determine statistically significant results. The following primers were used to quantify *PGM2* and *PGK1* mRNAs, as our positive control in different mutant strains. The primers used are as follows:

*PGM2*: Forward GGTGACTCCGTCGCAATTAT; Reverse: CGTCGAACAAAGCACAGAAA

*PGK1*: Forward ATGTCTTTATCTTCAAAGTT; Reverse: TTATTTCTTTTCGGATAAGA

### 4.5. Western Blot Analysis

Western blot analysis was used to investigate the protein content for Pgm2p-GFP fusion protein. Deleted mutant strains with GFP-tagged *PGM2* were grown in media treated with and without LiCl to investigate protein levels of *PGM2*. Protein extraction was performed as described by Szymanski et al. [75]. Bicinchoninic acid assay (BCA) was performed to estimate protein concentration as described by the manufacturer (Thermo Fisher^®^, Ottawa, ON, Canada). Equal amounts of total protein extract (50 μg) were loaded onto a 10% SDS-PAGE gel, run on Mini-PROTEAN Tetra cell electrophoresis apparatus system (Bio-Rad^®^). Proteins were transferred to a nitrocellulose 0.45 μm membrane via a Trans-Blot Semi-Dry Transfer (Bio-Rad^®^). Mouse monoclonal anti-GFP antibody and anti-Pgk1 (Santa Cruz^®^) were used to detect protein levels of Pgm2p-GFP and Pgk1p, respectively. Immunoblots were visualized with chemiluminescent substrates (Bio-Rad^®^) on a Vilber Lourmat gel doc Fusion FX5-XT (Vilber^®^). Densitometry analysis was carried out using the FUSION FX software (Vilber^®^, Collégien, France). Experiments were repeated at least three times. *T*-test analysis (*p*-value ≤ 0.05) was used to determine statistically significant results.

### 4.6. Genetic Interaction Analysis

To investigate Genetic interactions (GI) of our candidate genes *NAM7*, *PUS2*, and *RPL27B*, a SGA was performed in a 384 format as described previously [26,37,38] to make double mutants. GI is presumed when double deletion phenotype causes a more extreme phenotype from single deletion [76], suggesting the deleted genes to be functionally interacting in parallel pathways, causing synthetic sick or lethality. First, the candidate gene is deleted in Mat “α” mating type and then crossed to three sets (≈ 1000) of single mutant Mat “a” mating type. After a few rounds of selection, haploid double gene deletion mutants were selected. Colony size was used as a measure of fitness [37,38]. Colony size of double mutant strains, was compared to that of single mutant strains to detect synthetic sickness or synthetic lethal [34,77]. In brief, the size of individual colonies was measure by an automated software using the corresponding numbers of pixels and normalized to the average size of the colonies on that plate. Normalized colony sizes were then compared and size reductions of 50% to 95% were selected. This is a very conservative range resulting in the selection of the high confidence hits. This experiment was repeated three times. Only those hits that were observed in at least two repeats were considered as before [26,28].

For conditional SGA analysis, we grew our double mutant strain on media with a sub-inhibitory concentration of LiCl (3 mM, approximately 1/3 of the concentration used for strain sensitivity analysis) as a stress condition chemical [36] investigating genes that are expressed only under LiCl treatment [27,50].

For PSA analysis, we mated Mat “α” carrying over-expression plasmid of our query gene to entire library of single mutant Mat “a” similar to SGA procedure as mentioned above. In the last step, Mat “a” strains carrying over-expression plasmid are compared to that of Mat “a” carrying a control plasmid [36], both grown in media treated with LiCl. Possible GIs were selected based on gained fitness of mutants when over-expression plasmid of the candidate gene was introduced in the cells, showing compensation of sub-inhibitory effects of LiCl compared to relevant mutant showing sickness or lethality. In this manner we can propose a functional connection between these two genes [65,78].

### 4.7. Genetic Interaction Data Analysis

Yeast cell fitness is scored and measured based on colony size measurements as described previously [34,77]. We set our threshold at fitness to minimum of 30% reduction compared to control the mean average of control plate. Each experiment was repeated at least three times, and mutual hits that were observed at least in two repeats were selected. A list of hits then categorized based on their biological process and/or molecular function, with enriched GO terms using Gene Ontology Resources: http://geneontology.org/ (accessed on 18/11/2109), Genemania database http://genemania.org (accessed on 18/11/2019) and profiling of complex functionality http://webclu.bio.wzw.tum.de/profcom/start.php (accessed on 18/11/2019).

## 5. Conclusions

Here, we investigated LiCl sensitivity for different yeast gene deletion mutant strains, and, in doing so, we reported new activities for *NAM7*, *PUS2*, and *RPL27B* in the translation of structured mRNAs including that for *PGM2*. Our observations suggest that the translation of structured mRNAs seem to be more complicated than previously thought. Our results also indicate that LiCl sensitivity analysis can be utilized as an effective tool to discover new gene functions in the translation of structured mRNAs.

## Figures and Tables

**Figure 1 ijms-21-05730-f001:**
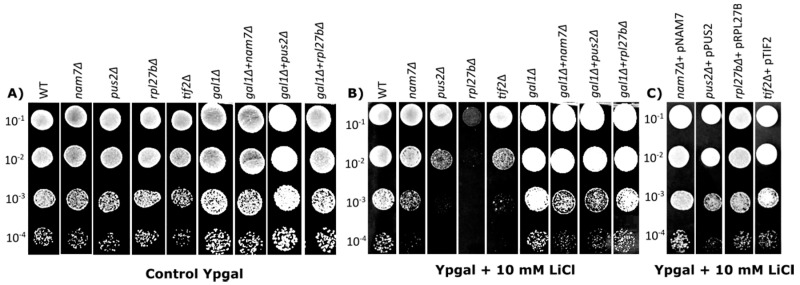
Lithium chloride (LiCl) sensitivity analysis for different yeast strains using spot test analysis. In (**A**–**C**) yeast cells were serially diluted as indicated (10^−1^ to 10^−4^) and spotted on Ypgal (Yeast extract, Peptone, and Galactose) media with or without LiCl (10 mM). In (**A**,**B**), *nam7Δ*, *pus2Δ*, and *rpl27bΔ* show less growth under LiCl treatment compared to wild type (WT). (**C**) Over-expression of target gene in their corresponding deletion mutants reverted cell sensitivity to LiCl (10 mM). Each experiment was repeated at least three times (*n* ≥ 3) with similar outcomes.

**Figure 2 ijms-21-05730-f002:**
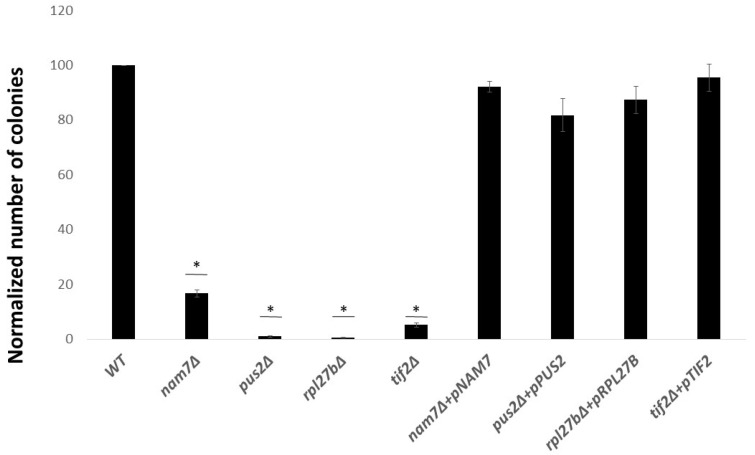
Quantitative analysis of LiCl sensitivity for different yeast strains. Average number of colonies formed for different yeast strains in the presence of LiCl (10 mM) were normalized to that for WT strain. *nam7Δ*, *pus2Δ*, and *rpl27bΔ* show less colonies under LiCl treatment. Over-expression of target genes in their corresponding deletion mutants reverted cell sensitivity to LiCl. * (*p*-value ≤ 0.05) represent statistically significant results compared to WT. Data represents the average from three independent experiments and error bars represent standard deviation.

**Figure 3 ijms-21-05730-f003:**
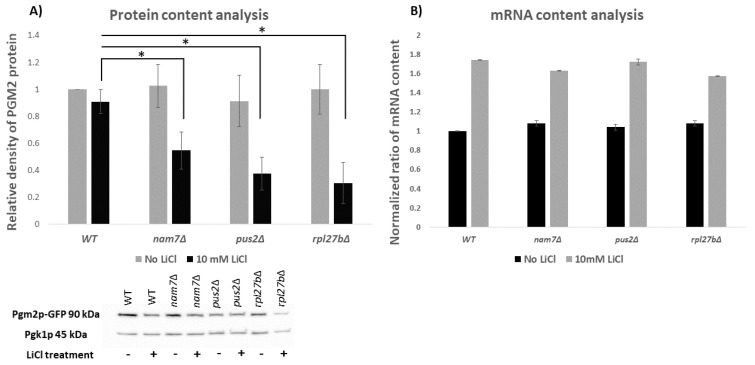
Protein and mRNA content analysis. (**A**) Protein content analysis of Pgm2p-GFP protein in deletion yeast strains for *nam7Δ*, *pus2Δ*, and *rpl27bΔ*. Western blot analysis was used to measure the protein content for Pgm2p-GFP protein in the absence or presence of LiCl (10 mM). Pgk1p was used as housekeeping gene and the values are normalized to that. (**B**) mRNA content analysis of *PGM2* in *nam7Δ*, *pus2Δ*, and *rpl27bΔ*. qRT-PCR was used to evaluate the content of *PGM2* mRNA in yeast gene deletion mutants related to WT strain and normalized to *PGK1* mRNA levels. Each experiment was repeated at least three times. * (*p*-value ≤ 0.05) represent statistically significant results compared to WT. Error bars represent standard deviation.

**Figure 4 ijms-21-05730-f004:**
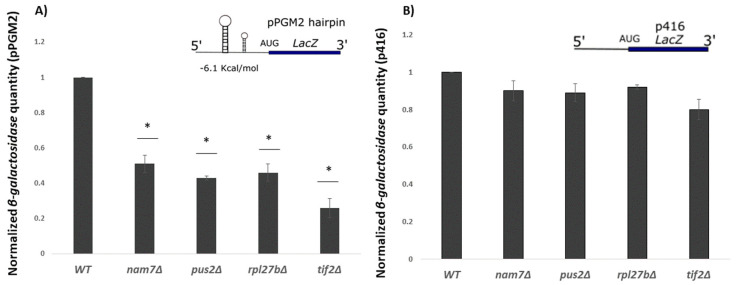
*β-galactosidase* expression analysis in different yeast strains. (**A**) Activities from *β-galactosidase* mRNAs that carry 5′-UTR of PGM2 mRNA (pPGM2) upstream of *LacZ* reporter was reduced in *nam7Δ*, *pus2Δ*, and *rpl27bΔ* strains. Strains carrying low complexity regions upstream of *LacZ* reporters p416 (**B**) did not show as significant reductions in *β-galactosidase* activity. Values are normalized to that for WT which resulted in average *β-galactosidase* values of 38.1 U and 407.5 U for pPGM2 and p416 constructs, respectively. Each experiment was repeated at least three times (*n* ≥ 3) and error bars represent standard deviation. * represents statistically significant results (*p*-value ≤ 0.05) compared to the WT. *t*-test analysis was used to compare differences. The insets represent schematic reporter mRNA structures.

**Figure 5 ijms-21-05730-f005:**
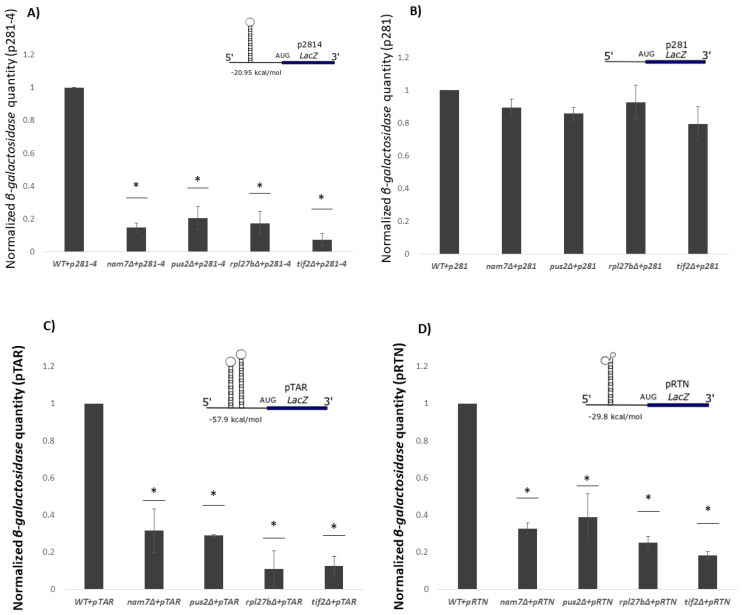
β-galactosidase expression analysis in different yeast strains. (**A**) Activities from *β-galactosidase* mRNAs that carry a strong hairpin structure (p281-4) upstream of a *LacZ* reporter were highly reduced in *nam7**Δ*, *pus2Δ*, and *hal1Δ* strains. (**B**) Our control plasmid (p281) with no inhibitory structure did not show reductions in *β-galactosidase* activity. pTAR (**C**) and pRTN (**D**) constructs contain the highly structured 5′-UTR of *HIV-tar1* and *FOAP-11* genes, respectively, in front of the *β-galactosidase* reporter mRNA showed significant reduction in mutant strains. Values are normalized to that for WT, which resulted in average *β-galactosidase* values of 14.1 U and 37.9 U for pRTN and p416 constructs, respectively. Each experiment was repeated at least three times (*n* ≥ 3) and error bars represent standard deviation. * represent statistically significant results (*p*-value ≤ 0.05) compared to the WT. *t*-test analysis (*p*-value ≤ 0.05) was used to compare differences. The insets represent schematic reporter mRNA structures.

**Figure 6 ijms-21-05730-f006:**
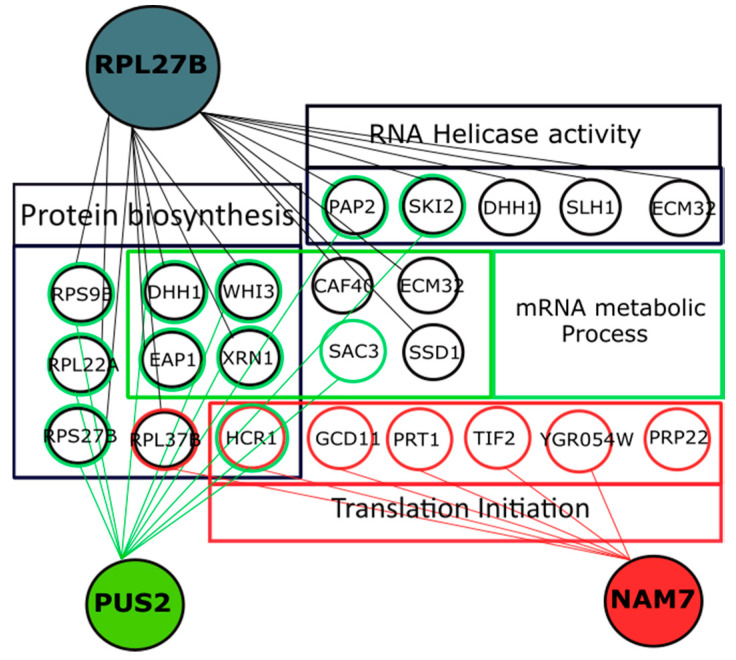
Negative genetic interactions (nGIs) for *NAM7*, *PUS2*, and *RPL27B*. Our data shows a cluster of interactors involved in protein biosynthesis pathway for *NAM7* (*p* = 1.6 × 10^−7^), *PUS2* (*p* = 4 × 10^−8^), and *RPL27B* (*p* = 1.2 × 10^−5^). *PAP2*, *SKI2*, *RPS9B*, *RPL37B*, *RPS27B*, *RPL22A*, *DHH1*, *EAP1*, *WHI3*, *HCR1*, and *XRN1* are mutual hits shared between two target genes. Circles represent genes, lines represent nGIs identified in this study.

**Figure 7 ijms-21-05730-f007:**
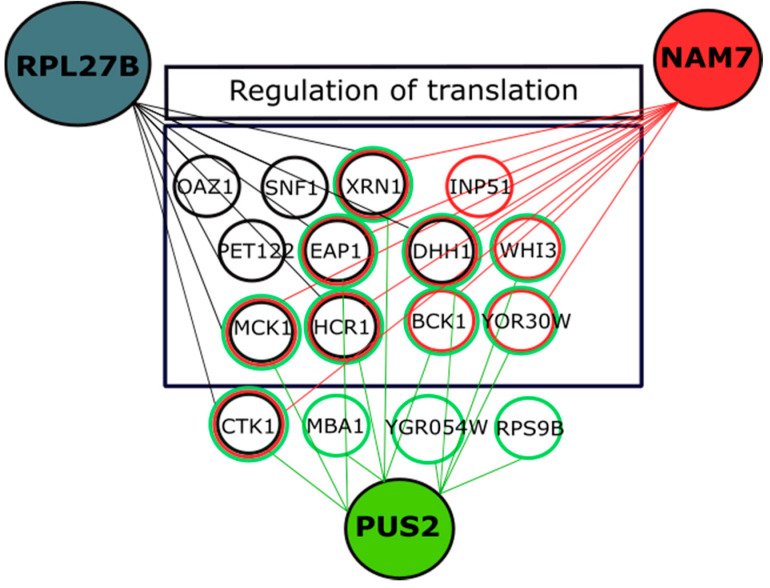
Conditional nGIs for *NAM7*, *PUS2*, and *RPL27B* under 3 mM concentration of LiCl. Our data shows a mutual Gene Ontology (GO) term enrichment for translation regulation genes for *NAM7* (*p* = 0.00037), PUS2 (*p* = 1.1 × 10^−6^), and RPL27B (*p* = 1 × 10^−6^). *BCK1*, *MCK1*, *CTK1*, *EAP1*, *XRN1*, and *DHH1* are mutual nGIs between the three quarry genes. Circles represent genes and lines represent nGIs identified in this study.

**Figure 8 ijms-21-05730-f008:**
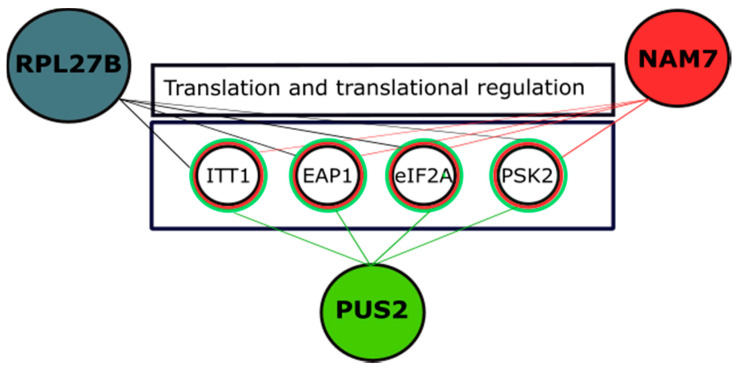
Over-expression of *NAM7*, *PUS2*, and *RPL27B* compensate for the sensitivity of *ITT1Δ*, *EAP1Δ*, *eIF2AΔ*, and *PSK2Δ* to 10 mM LiC. *EAP1* is known to be involved in negative regulation of translation via *DHH1* (RNA helicase) and PSK2 is involved in translation regulation of mRNAs with uORFs. eIF2A and *ITT1* participate in translation initiation and termination steps, respectively.

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
