# Peer review of "Lithium Chloride Sensitivity in Yeast and Regulation of Translation"

_ijms, 2020, doi:10.3390/ijms21165730_

Round 1
Reviewer 1 Report
This manuscript includes interesting data showing an impact of several genes related to translation and mRNA stability on the sensitivity to Li+ ions in yeast. The paper is clearly written, and results are of interest to scientific community. However, several issues have to be addressed before the manuscript can be accepted.
1) Strictly speaking, effect observed in a deletion strain from the yeast collection does not yet prove that the respectivephenotype is caused by this deletion. Some collection strains have additional mutations or other changes (e. g. loss of the [PIN+] prion). Thus, to prove that the effect is indeed due to a deletion, authors must show that the effect is compensated by reintroduction of the wild-type allele of the respective gene. This could be done by either transforming a deletion strain by a centromeric plasmid bearing the wild-type gene under its own promoter, or by a genetic cross with the isogenic wild type strain, followed by dissection and checking for the correlation be the phenotype and deletion in tetrads. Using of the overexpressor construct would not be sufficient, as authors themselves show the impact of overexpressors on the phenotype in other deletion strains.
2) Authors almost don't discuss the potential molecular mechanism that could explain the effects of the respective proteins on translation. More detailed discussion on this, separated from actual results would be useful. It appears that the manuscript provides sufficient evidence for such a discussion.
3) This would be interesting to compare synthetic effects observed by authors to data on the large scale screens for synthetic interactions, available for Saccharomyces Genome Database.
4) As correctly stated on Figure 3B, this experiment measures mRNA contents rather than transcription per se. However, discussion of these data in text is focused on transcription. this is not entirely accurate, because mRNA levels depend on both transcription and mRNA stability. This is not a formal point by far, as one of the genes detected by authors, NAM7, is involved in nonsense-mediated mRNA decay, and authors discuss the potential role of mRNA decay in the observed phenotype in connection to their data. However,alterations of mRNA decay would alter mRNA levels, that is not observed; thus, such an explanation is unlikely. I recommend authors to go through these statements, appearing at several places in the manuscript, and make them compatible to observed results.
5) Even though Li+ compounds are used in treatments of some diseases, terming them "drugs" as for example in the title of Fig. 2 is somewhat misleading.
Author Response
First reviewers’ comments to the authors:
1) Strictly speaking, effect observed in a deletion strain from the yeast collection does not yet prove that the respectivephenotype is caused by this deletion. Some collection strains have additional mutations or other changes (e. g. loss of the [PIN+] prion). Thus, to prove that the effect is indeed due to a deletion, authors must show that the effect is compensated by reintroduction of the wild-type allele of the respective gene. This could be done by either transforming a deletion strain by a centromeric plasmid bearing the wild-type gene under its own promoter, or by a genetic cross with the isogenic wild type strain, followed by dissection and checking for the correlation be the phenotype and deletion in tetrads. Using of the overexpressor construct would not be sufficient, as authors themselves show the impact of overexpressors on the phenotype in other deletion strains.
Our Response: We agree with this comment. Reintroduction of the deleted genes back into their corresponding deletion mutant strains was performed, using a modified SGA method, by mating the Mat “a” deletion mutant strain, carrying a G418 resistance gene marker in place of the deleted gene with a WT Mat “α” strain carrying a nourseothricin sulfate (clonNAT) resistance marker and selecting the target (G418 sensitive and clonNAT resistant) Mat “a” progeny, followed by PCR confirmation. The material and methods section, lines 378-383 are now modified to reflect this.
We observed that re-introduction of the deleted genes back into the genome compensated the observed LiCl sensitivity for the gene deletion strains. The results are shown in the Supporting Figure 1S, lines 765-771. Also, the results section, lines 108-110 are now modified to reflect this.
2) Authors almost don't discuss the potential molecular mechanism that could explain the effects of the respective proteins on translation. More detailed discussion on this, separated from actual results would be useful. It appears that the manuscript provides sufficient evidence for such a discussion.
Our response: We agree with this comment. We have now included a new discussion section. In this section we discuss the possible mechanisms by which NAM7, PUS2, and RPL27B can mediate the translation of structured mRNAs, lines 342-358.
3) This would be interesting to compare synthetic effects observed by authors to data on the large scale screens for synthetic interactions, available for Saccharomyces Genome Database.
Our response: We agree with this comment. The comparison of this data to the published negative genetic interaction in Saccharomyces Genome Database (SGD) shows an expected degree of overlap. For example, for NAM7, except for an observed nGI with YGR054W, other observed nGIs overlap with the reported interactions. There is no prior published information about conditional nGI or PSA analysis for LiCl. The results section, lines 264-267 are now modified to reflect this.
4) As correctly stated on Figure 3B, this experiment measures mRNA contents rather than transcription per se. However, discussion of these data in text is focused on transcription. this is not entirely accurate, because mRNA levels depend on both transcription and mRNA stability. This is not a formal point by far, as one of the genes detected by authors, NAM7, is involved in nonsense-mediated mRNA decay, and authors discuss the potential role of mRNA decay in the observed phenotype in connection to their data. However,alterations of mRNA decay would alter mRNA levels, that is not observed; thus, such an explanation is unlikely. I recommend authors to go through these statements, appearing at several places in the manuscript, and make them compatible to observed results.
Our response: We agree with this comment. The text is now modified accordingly. Lines 157 and 158 are modified accordingly
5) Even though Li+ compounds are used in treatments of some diseases, terming them "drugs" as for example in the title of Fig. 2 is somewhat misleading.
Our response: We agree with this comment. For clarity we replaced the word “drug” with “LiCl” or “chemical” were applicable. The text is modified in lines 101, 128, 135, 274, 376, 410, 411, and 473.
Reviewer 2 Report
This study uses standard yeast genomic tools to interrogate the genes required for the response to lithium chloride.
In its current form it is difficult to assess the significance of the manuscript and I encourage the authors to submit a substantial revision.
To go beyond the preliminary nature of the current work several elements should be included:
Perform a detailed survey of the literature on the effects of lithium and other salts on yeast gene-gene and gene-drug interactions
Provide detailed statistics so that a reader can interpret the colony size measurements
Discuss in a measured way, how the dosing of yeast with lithium might inform the treatment of bi-polar patients. The current discussion of this topic is quite thin.
Author Response
Second reviewers’ comments to the authors:
1) Perform a detailed survey of the literature on the effects of lithium and other salts on yeast gene-gene and gene-drug interactions.
Our response: We agree with this comment. We have now included the discussion of the effects of Lithium and other salts on yeast with new references in the Introduction section, lines 63-73 and 83-87.
2) Provide detailed statistics so that a reader can interpret the colony size measurements.
Our response: We agree with this comment. More detail is now added in lines 465-470.
3) Discuss in a measured way, how the dosing of yeast with lithium might inform the treatment of bi-polar patients. The current discussion of this topic is quite thin.
Our Response: We agree with this comment. More discussion is now added, lines 359-365.
Round 2
Reviewer 1 Report
The manuscript has been improved by revisions, however the description of mating experiment is somewhat unclear, making it hard to understand whenther or the major comment (Comment 1) of the previous review has been adequately addressed. Authors indicate that they mated the MATa yeast strain containing a deletion to the MATalpha partner (apparently containing a respective wild-type allele) and obtained cells which combined markers of both partners and were MATa. However, if authors indeed mated MATa strain with MATalpha partner, they should have obtained alpha/a diploids. To get MATa (haploid) cells, they had to go through sporulation and dissection. (However if such cells contained G418 resistance marker, they should have contained the deletion allele, not a wild type allele). The description needs to be clarified.
Moreover, if authors in fact obtained and analyzed an alpha/a diploid containing both deletion and wild-type alleles (as is my guess), this represents only the first step of analysis. This confirms that reintroduction of the wild-type allele recovers the phenotype, however it does not yet confirm that the phenotype was directly caused by a deletion. One example from literature is a control of polyglutamine toxicity by some deletions from yeast collections (see papers from M. Sherman, Y. Chernoff and S. Liebman labs). Some deletions antagonized aggregation and toxicity of polyQ constructs in yeast, however it turned out that in most cases, this occurred due to a loss of the yeast prion [PIN+] present in most strains of deletion collection but lost in some of them: in some strains, this loss was caused by a respective deletion, but in the other strains, was simply a coincidence, as transformation procedure involved in deletion construction may cause a prion loss with some frequency as well. In the latter case, reintroduction of wild-type allele through a genetic cross seemed to restore toxicity, however it turned out that this occurred because of reintroduction of the [PIN+] prion present in a wild-type strain. In fact, a significant number of anti-toxic deletions identified in the large scale screen by Giorgini lab turned out to be such false positives. To avoid such a possibility, it is important to sporulate and dissect diploids, and to show that spore clones bearing the deletion show the phenotype initially assigned to this deletion, while spore clones bearing the wild type allele don't show this phenotype (as requested in my initial comments). If authors have troubles with sporulation/dissection technique, they need to contact the yeast expert and/or check yeast manuals e. g. F. Sherman 2002 Getting started with yeast. Until such a dissection analysis is performed, one can not conclude that a deletion is indeed a cause for the observed phenotype therefore this comment is by no means formal.
If dissection represents a problem, authors can perform random spore analysis after sporulation, or transform an original haploid deletion strain with a single-copy or integrative plasmid containing the wild-type allele of the deleted gene under its normal endogenous promoter. This would also suffice.
Other comments are addressed, and the paper can be accepted if this remaining comment is properly addressed.
Author Response
Our Response: We agree with this comment. We have now included more details and 2 references for the procedure. In brief, for reintroduction of the WT gene, we used a modified version of SGA and random spore analysis. This approach was originally developed to study negative genetic interactions in a high throughout manner by allowing the construction of yeast double gene mutants in a fast and efficient manner, through mating, sporulation and selection steps (Tong et al, Science 2001). Since then it has been modified and used for other purposes and in other organisms by different labs including ours (e.g. Dixon et al, PNAS 2008; Yan et al, Nature Meth 2008; Alamgir et al, BMC Genomics 2008; Omidi et al, PLoS One 2014). The slide below briefly describes the procedure:
(figure provided in the attached file)
In this case we modified the selections to select for haploid MAT“a” progeny, resistant clonNat but sensitive to G418. As we often do, 5 colonies were selected, and the presence of the target WT gene was confirmed using PCR. Also we subjected all 5 confirmed colonies to LiCl sensitivity analysis and observed similar profiles. Additional information is added to the materials and methods section lines 386-394.

Reviewer 2 Report
The authors have addressed my prior concerns and the revised manuscript is much more accessible.
Although they do not interrogate essential genes in the work, I would strongly suggest that they cite our work that the protein product of the IPP1 gene, a cytoplasmic inorganic pyrophosphatase is specifically inhibited by NaF. Adding the two references PMID: 19332878 and PMID: 18622398 to the discussion would nicely complement their study and suggest further investigation into a detailed mechanistic study.
Author Response
Our Response: We agree with this comment. We have now included the two mentioned references in the discussion section lines 320-326.